# Transcriptomic Profiling of Equine and Viral Genes in Peripheral Blood Mononuclear Cells in Horses during Equine Herpesvirus 1 Infection

**DOI:** 10.3390/pathogens10010043

**Published:** 2021-01-07

**Authors:** Lila M. Zarski, Patty Sue D. Weber, Yao Lee, Gisela Soboll Hussey

**Affiliations:** 1Department of Pathobiology and Diagnostic Investigation, Michigan State University, East Lansing, MI 48824, USA; zarskili@msu.edu (L.M.Z.); windy77919@gmail.com (Y.L.); 2Department of Large Animal Clinical Sciences, Michigan State University, East Lansing, MI 48824, USA; weberp@msu.edu

**Keywords:** EHV-1, herpesvirus, horse, PBMC, transcriptomics, RNA sequencing, microRNA, gene expression

## Abstract

Equine herpesvirus 1 (EHV-1) affects horses worldwide and causes respiratory disease, abortions, and equine herpesvirus myeloencephalopathy (EHM). Following infection, a cell-associated viremia is established in the peripheral blood mononuclear cells (PBMCs). This viremia is essential for transport of EHV-1 to secondary infection sites where subsequent immunopathology results in diseases such as abortion or EHM. Because of the central role of PBMCs in EHV-1 pathogenesis, our goal was to establish a gene expression analysis of host and equine herpesvirus genes during EHV-1 viremia using RNA sequencing. When comparing transcriptomes of PBMCs during peak viremia to those prior to EHV-1 infection, we found 51 differentially expressed equine genes (48 upregulated and 3 downregulated). After gene ontology analysis, processes such as the interferon defense response, response to chemokines, the complement protein activation cascade, cell adhesion, and coagulation were overrepresented during viremia. Additionally, transcripts for EHV-1, EHV-2, and EHV-5 were identified in pre- and post-EHV-1-infection samples. Looking at micro RNAs (miRNAs), 278 known equine miRNAs and 855 potentially novel equine miRNAs were identified in addition to 57 and 41 potentially novel miRNAs that mapped to the EHV-2 and EHV-5 genomes, respectively. Of those, 1 EHV-5 and 4 equine miRNAs were differentially expressed in PBMCs during viremia. In conclusion, this work expands our current knowledge about the role of PBMCs during EHV-1 viremia and will inform the focus on future experiments to identify host and viral factors that contribute to clinical EHM.

## 1. Introduction

*Herpesviridae* are ubiquitous pathogens that infect most mammals, birds, and reptiles. They are enveloped, double-stranded DNA viruses with the trademark ability to establish life-long latency in their hosts. The latent genome packages itself within the cell nucleus as a circular extra-chromosomal episome of viral genomic DNA [1]. Latent infections are typically subclinical, though reactivation or reinfection from another host can produce active clinical disease. In horses, the most commonly described herpesviruses include *Equid alphaherpesvirus 1* (EHV-1), *Equid gammaherpesvirus 2* (EHV-2), *Equid alphaherpesvirus 4* (EHV-4), *and Equid gammaherpesvirus 5* (EHV-5) [2]. These viruses fall into two main subfamilies: *Alphaherpesvirinae* (EHV-1 and EHV-4) and *Gammaherpesvirinae* (EHV-2 and EHV-5). Alphaherpesviruses are known for their rapid lytic replication in many cell types [2]. Alphaherpesviruses typically (though not exclusively) establish latency in the sensory ganglia of their host; however, EHV-1 is also known to establish latency in lymphoid tissues [3,4,5,6,7]. On the other hand, gammaherpesviruses are more restrictive in their cell tropism and are known to establish latency primarily in lymphocytes as well as for their slow replication cycle [1,8].

In horses, alpha- and gammaherpesviruses are ubiquitous and have been reported to be detected in their respective tissues of latency in over 75% of healthy animals [2,4,8]. For equine gammaherpesviruses, the role of infection in clinical disease is not well understood. However there are reports showing an association with pulmonary fibrosis, pharyngitis, dermatitis, lymphoma, and other conditions [8]. In contrast, active infections with the equine alphaherpesviruses EHV-1 and EHV-4 are primary causes of acute respiratory disease in younger animals, which is a major contributor to loss of training [9]. In addition, EHV-1 also can cause late-term abortion, neonatal death, ocular disease, or the neurologic disease equine herpesvirus myeloencephalopathy (EHM) in up to 10% of infected cases [9]. This is in contrast to EHV-4 infection, which typically stays restricted to the respiratory tract and does not cause abortions or EHM [10].

The key reason for the difference in secondary clinical disease manifestations between EHV-1 and EHV-4 is the fact that EHV-1 establishes viremia, which is essential for transporting the virus from the respiratory tract to the secondary sites of infection. In contrast, viremia is not known to be a central feature of EHV-4 disease pathogenesis. EHV-1 establishes viremia in peripheral blood mononuclear cells (PBMCs) shortly after initial infection of the respiratory epithelium. Within the population of PBMCs, monocytes, T cells, and B cells have been shown to become infected with EHV-1, and there are conflicting reports on which of these subpopulations the virus prefers [11,12,13,14,15]. Unlike the lytic life cycle of infected epithelial cells in which full transcription and replication of virions destroys cells and produces free virus, infection in PBMCs is characterized by a restricted viral gene expression, which delays viral replication until the cell has made contact with the vascular endothelium [16,17,18]. This allows the virus to effectively evade immune surveillance within the PBMCs while being transported to the secondary sites of infection. Furthermore, it is likely that viremia is an important step in the establishment of latency of EHV-1 in lymphoid tissues [7].

Thus, the period of viremia is a critical prerequisite for the development of secondary diseases such as EHM and abortions. It has been shown that EHM generally develops during the end of viremia and that a longer duration and higher magnitude of viremia contribute to the likelihood of EHM [19]. It is presumed that the prolonged exposure to infected PBMCs increases the potential of viral transfer to the vascular endothelium. In addition to the necessity of viral transfer to the vascular endothelium, perhaps the most critical factor affecting vascular damage and, subsequently, the clinical outcome is the immunopathology that occurs at this site. Because PBMCs are a robust immune and inflammatory cell population in the vasculature, as well as carriers of EHV-1 virus, they are an important focus of investigation of EHV-1-associated secondary diseases.

Despite the importance of PBMCs for the development of secondary EHV-1-associated diseases, limited information is available about the host or viral gene expression in these cells during this period. At this point, various cytokines such as interleukin-10, interferon gamma (IFNγ), and tumor necrosis factor beta have been shown to be induced in the blood of horses during EHV-1 viremia [20,21]. In vitro, EHV-1 induces expression of interferons, cytokines, and chemokines in both epithelial cells and PBMCs [22,23]. However, a complete profile of host and viral messenger RNA (mRNA) expression has not yet been performed.

In addition, micro RNAs (miRNAs) may play a role in the regulation of host and viral gene expression in PBMCs during viremia. MiRNAs are small (~22 bases) RNA molecules that block the translation of their target coding mRNAs [24]. It is becoming clear that host and viral miRNAs play an important role in the evasion of immune detection as a response to EHV-1 infection. Cellular (host) miRNAs have been shown to inhibit viral genome replication for herpes simplex virus 1 (HSV-1), which likely contributes to the virus’ ability to persist in the cell undetected [25]. Virally encoded miRNAs have been identified in several herpesviruses and play an important role in viral persistence in cells by downregulating host immune responses [26]. Currently, the role of miRNAs is unknown for EHV-1 replication.

The goal of this study was to use RNA sequencing to analyze the mRNA and miRNA transcriptome of equine PBMCs before and during EHV-1 viremia as an unbiased and comprehensive approach to gene expression analysis. Our rationale was that this approach can be used to reveal mechanisms involved in infection of the vascular endothelium and facilitate understanding of the events that contribute to EHV-1 secondary disease. This information is essential to aid in the development of prophylactics or treatments against secondary EHV-1 diseases and will inform the focus on future experiments to identify host and viral factors that contribute to EHM and abortions.

## 2. Results

### 2.1. Clinical Disease and Viremia

All horses were free from clinical signs of respiratory disease and had normal body temperatures prior to infection with EHV-1. As described previously, all horses developed fevers, shed virus in nasal secretions, and seroconverted following EHV-1 inoculation, indicating successful challenge infection (Figure 1A,C) [27]. As expected, viral DNA was more prominent in nasal secretions than in PBMCs. One horse (horse 6) developed severe EHM and was euthanized day 11 post-challenge. PBMC samples were analyzed for viremia for 10 days following challenge by quantitative real-time PCR (qPCR). The day of peak viremia was identified for each horse and occurred between days 5 and 8 post-infection (p.i.) (Figure 1B) [27].

### 2.2. Horse mRNA Sequencing and Differential Gene Expression

The mapping summary statistics is shown in Table 1 and describe the number of total reads and uniquely mapped reads. The read depth was 43,480,084 total reads on average per sample, and an average of 80.3% of reads uniquely mapped to the equine genome.

The principal component analysis (PCA) of regularized log-transformed read-count data shows a clustering of samples based on timepoint (pre-infection vs. post-infection) (Figure 2A). When comparing pre-infection with post-infection samples, we found a total of 3226 differentially expressed genes (DEGs) with an adjusted *p* value (padj) < 0.05 (Appendix A). Due to the high number of DEGs and in order to highlight the most relevant biological processes in downstream enrichment analyses, we set the significance threshold at padj < 0.05 and log2 fold change greater than 3 or less than −3. At this threshold, there were 51 DEGs, 48 of which were significantly upregulated and 3 genes that were significantly downregulated (Figure 2B).

Functional information for available genes from the *Homo sapiens* uniprotKB database revealed numerous genes involved in various aspects of inflammation. The majority of upregulated genes belonged to the interferon pathway (such as *DDX60*, *MX2*, *MX1*, *GBP2*, *IFIT3*, *IFI44*, *OAS1*, *OASL*, *OAS3*, *TRIM22*, *OAS2*, *IFIT5*, *IFI6*, and *IRF7*). In addition, many chemotactic genes (*CXCL9*, *CXCL10*, *CXCL11*, and *CCL8*) as well as genes involved in the complement system (*C1R*, *C3AR1*, and *SERPING1*) were upregulated, as well as other genes which are shown in Table A1. The three downregulated genes included fibronectin (*FN1*), which encodes an adhesion molecule, as well as *DEFB1*, the gene that encodes the antimicrobial protein beta defensin 1, and the gene *FAM71A*.

### 2.3. Gene Ontology (GO) Overrepresentation

Due to the low number of downregulated genes, the gene ontology (GO) overrepresentation analysis focused on processes involved in the upregulated gene list. The gene ontology overrepresentation analysis was performed on the 48 upregulated genes and 150 total GO terms for biological processes were enriched (Appendix A). After summarizing the list with a tool that summarizes lists of GO terms based on semantic similarity (REVIGO), 18 non-redundant enriched GO terms remained (Figure 3A). The most significantly enriched process was defense response to virus (GO:0051607), which also included the most genes (16) from our gene list. The other top significantly enriched processes were negative regulation of viral genome replication (GO:0045071), regulation of nuclease activity (GO:0043950), positive regulation of cyclic adenosine monophosphate (cAMP)-mediated signaling (GO:0043950), and cellular response to chemokines (GO:1990869).

Genes associated with the top nine most significantly enriched GO terms are visualized in Figure 3B. These genes and processes cluster into three general groups. The largest group contains genes and processes involved in the defense response to viruses, negative regulation of viral genome replication, and regulation of nuclease activity and includes *DDX60*, *MX2*, *MX1*, *GBP2*, *IFIT3*, *IFI44*, *OAS1*, *OASL*, *OAS3*, *TRIM22*, *OAS2*, *IFIT5*, *IFI6*, and *IRF7*. The genes for chemokines *CCL8*, *CXCL9*, *CXCL10*, and *CXCL11* are involved with a second cluster of biological processes including cellular response to chemokines, positive regelation of cAMP-mediated signaling, and positive regulation of release of sequestered calcium ion into cytosol. The third main cluster includes three genes associated with the complement system, *C1R*, *SERPING1*, and *C3AR1*, which are involved with the enriched terms regulation of protein activation cascade, protein activation cascade, and regulation of protein maturation.

### 2.4. In Silico Cell Sorting

Average cell fractions between pre- and post-infection samples and the averages ± SEM for each cell type are shown in Table 2. Following challenge infection during peak viremia, there was a significant increase in γδ T cells (*p* < 0.05) and M1 polarized macrophages (*p* < 0.1). Additionally, there was a significant reduction in CD8+ T cells (*p* < 0.05), plasma cells (*p* < 0.1), and M0 macrophages (*p* < 0.1) post-infection.

Relative cell percentages for 22 different leukocyte subpopulations in individual horses pre-and post-infection can be seen in Figure 4. Overall, the most abundant cell type identified in all samples was naïve B cells, with around 40% of the estimated cell fraction, followed by follicular T helper cells with ~17%. Interestingly, horse #6, who was the only horse exhibiting clinical EHM, was also the only horse showing an increase in naïve B cells and T follicular helper cells and one of two horses exhibiting visible increases in the percentage of T regulatory cells.

### 2.5. Viral mRNA Sequencing

Normalized read counts in transcripts per million (TPM) for EHV-1, EHV-2, and EHV-5 were identified and are shown in Appendix A. No reads mapped to the EHV-4 genome. Read coverage along the EHV-1 genome shows a low level of transcription of viral reads in 5/7 of the horses prior to EHV-1 challenge (Figure 5). As expected, during EHV-1 viremia post-challenge, read coverage of the EHV-1 genome increased and EHV-1 transcripts were present in all samples (Figure 5). Additionally, the horses with the highest levels of viremia corresponded to the samples with the most EHV-1 transcription post-challenge (Figure 1B; Figure 5). TPM values for each gene were then averaged and the most abundant EHV-1 genes expressed post-infection were ORF25, ORF34, and ORF75 (Figure 6A). The products of these genes include a capsid protein (ORF25) and a protein involved in the early step of virus egress (ORF34). ORF75 encodes a membrane protein presumed to be involved in the virulence of certain EHV-1 strains [29,30]. The most abundant EHV-1 genes pre-infection were ORF59, ORF25, and ORF58 (Figure 6A). ORF59 encodes an early protein involved in viral growth and ORF58 encodes a nuclear non-structural protein [30,31]. Interestingly, three genes, namely ORF59, ORF41 (encodes a membrane protein), and ORF55 (encodes a tegument protein), had higher levels of transcription in pre-infection PBMCs compared to PBMCs during viremia. This could indicate a role for these genes in EHV-1 persistence in PBMCs during clinical latency.

Additionally, transcription of the equine gammaherpesvirus (EHV-2 and EHV-5) genes was present in PBMCs prior to and after EHV-1 challenge (Figure 6B,C). No apparent differences were observed in TPM of EHV-2 or EHV-5 genes in response to EHV-1 infection.

### 2.6. Identification of miRNAs

MiRDeep2, a software tool for miRNA mapping and identification, identified 278 known mature equine miRNAs amongst the pooled samples. Furthermore, we identified 903 total novel miRNAs with an miRDeep2 score >1 (Appendix A). Of these 903 novel miRNAs, 855 mapped to the equine genome, 57 mapped to the EHV-2 genome, and 41 mapped to the EHV-5 genome. The EHV-2 and EHV-5 miRNAs appeared to be relatively abundant given the low levels of their viral mRNA transcripts detected. The most abundant miRNA for EHV-2 had a total read count of 15,671, and for EHV-5, a count of 20,551. For EHV-2, the miRNAs clustered around three general regions on the genome: 38–44 and 176–182 kb on the plus strand, and 124–127 kb on the minus strand. For EHV-5, the miRNAs clustered around two general regions: 36–43 kb on the plus strand and 126–127 kb on the minus strand (Appendix A). Interestingly, no miRNAs were identified that mapped to either of the equine alpha herpesviruses, EHV-1 or EHV-4 (Appendix A).

### 2.7. Differential Expression of miRNAs

Novel miRNAs with an miRDeep2 cutoff > 1 were added to the list of known mi-RNAs for each sample and a quantification was performed to identify differentially expressed miRNAs in horses before and after EHV-1 infection. There was an average of 18,165,757 reads per sample with an average mapping of 49.7% (Table A2).

Interestingly, PCA plot analysis of these counts indicated that more variation occurred between individual horses rather than within horses as a result of infection (Figure 7A). Furthermore, there was no significant correlation between levels of viremia or EHM and miRNA expression.

However, differential expression analysis between pre-and post-infection samples revealed five total miRNAs that were differentially expressed in horses during viremia when compared to pre-infection levels (Table 3).

Three of these miRNAs were upregulated in response to infection, and two were downregulated. The upregulated miRNAs included eca-miR-9104, eca-miR-2483, and eca-miR-652 (human ortholog hsa-miR-652-3p) (Figure 7B–D). Predicted target genes that were also differentially regulated in our samples included *TNRC6A*, *NPTN*, *KPNA1*, and *TP53*. *TNRC6A* is a regulatory gene involved in gene silencing and gene expression regulation. *NPTN* plays an important role in cell adhesion molecule binding and type 1 fibroblast and neurite growth. *KPNA1* has been shown to play a role for nuclear import of proteins, including herpesvirus proteins [32], and *TP53* plays a role in the regulation of apoptosis and cell cycle regulation. Finally, in humans, *hsa-miR-652-3p* has been shown to interact with the endothelial repair gene *CCND2* and contributes to endothelial cell damage [33]. While the downregulated miRNA list did not include any known equine mi-RNAs, we did identify a “novel” equine miRNA with the human ortholog *hsa-miR-6852-5p* to be downregulated in response to infection (Figure 7E). Interestingly, *hsa-miR-6852-5p* is predicted to target a number of the differentially expressed genes in our study. These include *MX2*, *OAS2,* and *OAS3,* as well as *IFIT5*, which all belong to the interferon-stimulated genes and are important components in the defense of herpesviruses. Further differentially regulated targets include *CCL8,* which is an important chemokine, and a number of genes that are involved in immune defense, apoptosis, cortisol metabolism, and cell adhesion (*BCL2LI4*, *HSD11B1*, *MPZ*, *TBM2*, *DEFB1,* and *FN1*). In humans, *hsa-miR-6852-5p* has also been shown to induce cell cycle arrest and necrosis [34]. Perhaps even more intriguing is the fact that *hsa-miR-6852-5p* is predicted to bind tetraspanins (*TSPN9*, *TSPN11*, *TSPN14*, *TSPN15*, *TSPN17*, *TSPN18*, and *TSPN31*) as well as many collagens. A recent paper comparing horses that developed clinical EHM with horses that did not, identified a mutation in *TSPAN9* in horses with EHM in a genome-wide association study [35]. *TSPAN9* is expressed in endothelial cells and platelets and stabilizes the collagen receptor in platelet microdomains [36]. While the collagen receptor plays only a minor role in hemostasis, it is important in arterial thrombosis, ischemic stroke, and maintaining vascular integrity during inflammation. Interestingly, we have recently found that plasma collected from horses with clinical EHM responded with decreased aggregation in response to collagen (unpublished data). Furthermore, absence of *TSPAN9* in mice reduced collagen-induced activation and secretion [37]. Finally, a novel EHV-2 miRNA was also found to be downregulated in our samples (Figure 7F).

## 3. Discussion

PBMCs are an important tissue involved in the pathogenesis of EHV-1 disease both as active immune cells in addition to being the site of EHV-1 viremia and potentially latency. In this study, we sought to characterize the host and viral transcriptome of equine PBMCs prior to and during acute equine herpesvirus 1 infection. In addition, we identified miRNA expression in these cells, as they likely contribute to the tight regulation of and switch between lytic and latent infection. Using next-generation RNA sequencing (RNA seq), we were able to build an unbiased profile of equine and viral gene expression in PBMCs and determined which genes were modulated during EHV-1 viremia.

In total, 51 host genes were significantly differentially regulated in the PBMCs of horses collected during peak EHV-1 viremia when compared to PBMCs collected pre-challenge infection. As expected, we found that numerous genes involved in the interferon pathway were upregulated during EHV-1 viremia. This included many classified as interferon-stimulated genes (ISGs), including *OAS1*, *-2*, and *-3*, *OASL*, *MX1* and -*2*, *IFIT3*, *IFIT5*, *IFI6*, *TRIM22*, *GBP1*, and *DDX60* as well as the equine miRNA *miR-6852-5p* targeting a number of these genes.

The type I interferon response is generally thought to occur in three phases: 1) the stimulation of pattern recognition receptors and activation of tissue factors, including *IRF3* and *IRF7*; 2) induction of interferon alpha and interferon beta expression; and 3) the continued amplification and expression of ISGs through Janus Kinase/Signal Transducer and Activator of Transcription (JAK/STAT) signaling [38,39]. Similar to the observed upregulation of ISGs, we also observed the upregulation of the transcription factor interferon regulatory factor 7 (*IRF7*). This gene is stimulated upon pattern recognition receptor (PRR) stimulation and is involved in the induction of the type I interferon response [38,40]. The type I interferon response pathway is a crucial aspect of antiviral innate immunity and is considered an important response against herpesviruses [38]. In humans, deficiencies in the responsiveness to type I interferons and the inability to induce ISGs result in death from viral infections. In one case study, an infant with a homozygous mutation in the *STAT1* gene succumbed to HSV-1 infection with uncontrolled encephalitis [41]. PBMCs from patients with atopic dermatitis complicated by eczema herpeticum show significantly lower type I interferon response to ex-vivo HSV-1 infection compared to PBMCs from patients without eczema herpeticum [42]. In addition, this study found that the regulating genes *IRF3* and *IRF7* were also downregulated in these patients [42]. Together, these data suggest that the type I interferon response is important for protection against disease caused by alphaherpesviruses. Less is known about the importance of the type I interferon response in PBMCs from horses. In previous studies, PBMCs have been found to produce type I interferon in response to EHV-1 infection in vitro [22,23]. Similarly, in other equine cell types, type I interferon has been shown to be upregulated in response to EHV-1 infection in vitro, including respiratory epithelial cells and endothelial cells [43,44,45,46,47,48]. In our study, it was interesting to notice that the interferon alpha or interferon beta genes themselves were not upregulated, but rather the ISGs were. This was also the case in a gene expression study of human PBMCs in response to a variety of viral infections, where the authors found multiple interferon-inducible genes yet no detectable interferons [49]. Type 1 interferons are known to be quickly and transiently expressed, and so timing likely plays a role in detection of these genes. In epithelial cells, type I interferon proteins have been shown to be detectable as early as 10 h post-EHV-1 inoculation in vitro and remained detectable up to 72 h [46]. It might also be considered that the persistence of the antiviral transcriptome is maintained without further induction of type 1 interferons. In fact, it has been shown that ISGs are expressed in the absence of interferon alpha/beta signaling through *IRF7*-mediated pathways [50]. This indicates that future studies interested in the interferon response may consider investigating downstream molecules of this pathway, such as the numerous ISGs we found here.

Another major group of genes upregulated in PBMCs during EHV-1 viremia were chemokines (*CXCL9*, *CXCL10*, *CXCL11*, and *CCL8*). More specifically, we found upregulation of chemokines to be important for induction of cell-mediated immunity and recruitment of T cells, monocytes, and natural killer (NK) cells. *CXCL9*, *CXCL10*, and *CXCL11* are related chemokines that are well known for their chemotactic activity, as their receptor (CXCR3) can be found on type 1 CD4+ T helper cells, CD8+ cytotoxic lymphocytes, and NK cells [51]. Previous work has also found strong induction of these chemokines in PBMCs in response to EHV-1 infection in vitro [22]. In addition, these chemokines have been shown to be stimulated in epithelial and endothelial cell cultures following EHV-1 inoculation [52,53]. The gene encoding *CCL8* (also known as monocyte chemotactic protein-2; MCP-2) was also found to be upregulated in PBMCs during EHV-1 viremia in our study in addition to *miR-6852-5p*, which is predicted to target *CCL8*. This protein, as its name suggests, is chemotactic for monocytes, as well as NK cells, T cells, eosinophils, and basophils [54]. Not much is known about the role of *CCL8* during EHV-1 infection; however, it has been shown to be upregulated in endothelial cell cultures [53]. The CXCR3 ligands, as well as *CCL8*, are known to be induced by type II interferon (interferon gamma) [51,54]. Interferon gamma is secreted by T cells and NK cells and is considered a correlate of an active cell-mediated immune response [55,56]. Cellular immunity, particularly CD8+ T cells, is considered to be an important correlate of protection from EHV-1 [19,57,58]. Re-stimulation assays have found that T-cells in horses infected with EHV-1 increase interferon gamma secretion, indicative of activated T cell responses [19,59,60,61]. The fact that all three of the CXCR3 ligands were upregulated in our samples, along with previous work, suggests that these chemokines are consistently stimulated during EHV-1 infection and that T cell activation and recruitment play a role during EHV-1 infection and possibly in the protection from EHM and EHV-1 abortions. However, the exact role of these chemokines in the pathogenesis of EHV-1-mediated disease remains unknown.

A third cluster of genes with related functions in protein activation cascades were upregulated in PBMCs during EHV-1 viremia. The complement cascade is a pathway of the innate immune system that involves a series of protein activation and cleavages that enhance antibody-mediated clearance of pathogens. We found two components of this system (*C1R* and *C3AR1*) as well as an inhibitor to this cascade (*SERPIN1*) to all be upregulated during infection in addition to the equine *miR-6852-5p*, which is predicted to target *C1R* ligand. EHV-1 and other herpesviruses are known to evade the complement system by “hiding” antigens within the PBMCs [62,63]. However, components of the complement system have also been shown to play a role in adaptive immunity through B-cell- and T-cell-related functions. The complement receptor encoded by *C1R* has been shown to have a central role in B cell responses [64]. In humans, about 15% of T cells express *C1R*, and this molecule is upregulated in CD4+ and CD8+ T cells upon activation [65,66,67]. Similarly, *C3AR1* is not expressed on naïve T-cells, but it is induced upon activation [68]. While these characteristics have not been described in equine lymphocytes, it is possible the differential regulation of these genes during EHV-1 infection corresponds with activation of lymphocytes. Additionally, the proteins involved in the complement cascade have also been shown to be associated with the coagulation cascade [69]. For example, the C1 inhibitor (*SERPIN1*) not only regulates the complement cascade but has also been shown to inactivate coagulation factors [70]. This is of particular interest because the coagulation cascade is known to be induced in horses during EHV-1 viremia, and it is thought that this is a major contributor to endothelial damage and subsequent immunopathology, causing secondary disease such as EHM [71,72,73,74]. Regulation of hemostasis is critical during health and disease with a tightly regulated balance between anti-coagulation and pro-coagulation factors. In a healthy individual, the body holds this balance towards a slight anti-coagulation state. However, in times of disease or vascular damage, coagulation factors are released to form a protective fibrin clot. These factors are tightly regulated so as to prevent clotting disorders [69,75]. Therefore, it is not surprising that we see gene expression for components (*C1R* and *C3AR1*) as well as regulators (*SERPIN1*) of the complement and coagulation cascades.

Upon infection with EHV-1, we observed a change in the cell fraction percent of certain cell populations. We saw a decrease in M0 macrophages and an increase in M1 macrophages, which indicates activation and polarization of macrophages to an M1 phenotype. This phenotype is believed to correspond with the antiviral and anti-inflammatory macrophage phenotype [76]. Additionally, there was an increase in γδ T cells. γδ T cells are known for their expression of IL-17, and while they do express a T cell receptor, they can also be stimulated via cytokines, making them similar to innate lymphoid cells [77]. During viremia, EHV-1 infects monocytes, T cells, and B cells; however, only between 1 to 10 out of 10^7^ PBMCs are estimated to be infected with EHV-1. Therefore, it is unlikely that drastic cell population changes are a result of EHV-1 infection of a certain subtype [11,12,13,14,15,78]. EHV-1 infection is marked by lymphopenia, and previous reports attributed this specifically to T cells [79,80]. This was confirmed by Charan et al. [80], who showed that post-EHV-1 infection autologous sera collected from horses contained increased levels of transforming growth factor beta and caused both non-specific and EHV-1-specific suppression of T cell responses. Furthermore, we have previously shown suppression of both CD4+ and CD8+ T cell responses in ponies on day 7 post-infection with EHV-1 [20]. In the present study, we found a decrease in CD8+ T cells during infection. CD4+ T cells were also decreased, although this decrease was not statistically significant. It is known that CD4+ type 1 helper T cells and CD8+ T cells are crucial for protection against EHV-1 disease, and the decrease in circulating PBMC population could be explained by the recruitment of these cells towards sites of infection, i.e., the nasal epithelial tissue and secondary sites of infection [19,57,58]. These results also correspond with the observed increases in the chemokine genes *CXCL9*, *CXCL10*, *CXCL11*, and *CCL8* during peak viremia. Similarly, the reduction in plasma B cells we observed may be a sign of these cells being recruited out of circulation to the site of primary infection. We estimated cell populations using CIBERSORTx, a new tool for digital cytometry that allows for cell population estimations of heterogenous populations based on bulk RNA sequencing data [28]. It must be noted that the results from this in silico analysis should be interpreted with caution, as the estimation of cell fractions was computed using a reference file based on the transcriptome of human leukocytes that Newman et al. developed [28]. In order to generate a more accurate estimation, a reference should be generated based on the transcriptome of classically sorted equine cells. Alternatively, the expensive technology of single-cell RNA sequencing could be used to characterize gene expression in individual cells and to more precisely estimate populations.

While the majority of the PBMC transcriptome belongs to host genes, it was another goal of ours to identify which EHV-1 viral transcripts were present in the samples. It was interesting to see that many EHV-1 transcripts were identified in 5 of 7 horses prior to EHV-1 challenge, despite horses being negative for viral genome using qPCR. However, the depth of sequencing of our samples was approximately 43 million reads per sample, so we could identify low levels of viral transcription. It is possible that this depth of next-generation sequencing is more sensitive than our qPCR assay, and it is also possible that viral transcripts are more abundant in samples than latent genomic DNA. While the trigeminal ganglia is the trademark site for EHV-1 latency, many studies have reported EHV-1 latency in lymphoid tissues and PBMCs as well as in additional ganglia and lymphoid tissues [3,4,5,6,7]. It is presumed that most horses become infected with EHV-1 at a very young age (days to weeks old) [81,82,83,84]. While the horses used in our study were seronegative for EHV-1 serum virus neutralization (VN) antibodies prior to challenge, it is presumed that they still may have previously been exposed to the virus and established latency. We propose the transcription of EHV-1 found in the pre-infection PBMCs is from a previously acquired but clinically “latent” virus. An important distinction must be made between clinical latency—the period of time after an infection when the host no longer has the disease—and cellular latency—when the viral genome is present within the cell but does not produce progeny virions [1]. Cellular latency is often characterized by a restricted gene expression pattern and a characteristic latency-associated transcript. This transcript has not yet been identified for EHV-1, and we are unable to speculate whether the transcription observed in our pre-challenge samples represents true cellular latency, an “arrested” state of virus replication, or a mixture of lytic and latent gene expression patterns in different cells. However, we identified three genes that exhibited higher average expression in the pre-challenge samples compared to the post-challenge samples. These included ORF59, ORF41, and ORF55. ORF59 was also the most abundant transcript overall in the pre-challenge samples. This gene encodes an early protein involved in viral growth and has been found to be essential for EHV-1 replication in culture [31]. Interestingly, ORF59 is one of a few genes without a positional homolog in HSV-1 or HSV-2. The homolog exists in other *Varicelloviruses* including Varicella-Zoster virus (VZV) and pseudorabies virus (PRV), and while it is essential for PRV replication, it has been found to be non-essential for VZV replication in culture [85,86]. The ORF25 gene was found to be relatively highly expressed in both pre- and post-challenge samples. This gene encodes a capsid protein. While not much is known about the role of this protein in EHV-1 infection, the VZV homolog (ORF23) has been found to be important for infection and lesion formation on human skin xenografts in a murine model of VZV pathogenesis [87]. During viremia, we also found ORF75 to be one of the most abundantly expressed genes. This gene is one of six genes that have been deleted from the KyA laboratory strain of EHV-1. KyA has been found to be attenuated and avirulent in horses and does not cause detectable viremia [29]. Creation of an ORF75 deletion mutant has identified that this gene is not essential for EHV-1 replication and does not alter the clinical virulence in vivo [88,89]. However, the complete function of ORF75 remains unknown.

In addition to transcription of EHV-1 genes, we found transcripts from both equine gammaherpesviruses, EHV-2 and EHV-5. This is unsurprising given that the site of persistence for these viruses is B cells [90,91]. Our results are also consistent with other reports that show that detection of EHV-2 and EHV-5 is common in PBMCs, while EHV-1 is predominantly detected during the viremic phase of acute EHV-1 infection [92]. Here, we found EHV-2 transcripts in samples of 4/7 horses pre- and post-EHV-1 challenge. For EHV-5, transcripts were detected in 5/7 pre-EHV-1 infection and in 3/7 post-EHV-1 infection samples, respectively. In a previous study of EHV-1-challenged horses, we found EHV-5 in PBMCs of 76% of horses prior to challenge infection and 52% seven and ten days post-EHV-1 challenge infection [93]. It is possible that the rate of EHV-5 detection drops below the detectable limit in PBMCs during EHV-1 viremia due to increased immune activity of the circulating lymphocytes. Studies linking EHV-5 and equine lymphoma point to an important role of lymphocytes in the control of EHV-5 persistence and re-activation [94,95]. However, both EHV-2 and EHV-5 transcripts were observed during acute EHV-1 infection. It is therefore likely that the gammaherpesviruses are resilient and persistent in this site of latency, even in the face of systemic immunity.

Notably, no miRNAs mapped to the EHV-1 genome prior to or during EHV-1 viremia. However, we identified 98 total miRNAs mapping to the two gammaherpesvirus genomes. MiRNAs are known to be involved in herpesvirus biology and the establishment of latency [26]. The differences we observed in miRNA between the gammaherpesviruses EHV-2 and EHV-5 and the alphaherpesvirus EHV-1 may be explained by the differences in these subfamilies’ latency tropism. The classical theory of herpesvirus biology describes sensory ganglia as the primary target for latency of alphaherpesviruses and lymphocytes as the primary latency site for gammaherpesviruses [1]. While several reports have identified lymphoid tissues as a site of latency for EHV-1, genome detection in circulating PBMC via qPCR is often negative in non-clinically affected animals [3,4,5,6,7,96]. In contrast, EHV-2 and EHV-5 can be routinely detected in PBMC samples of healthy animals [96,97]. This indicates that these two subfamilies employ differing transcription strategies in latency and persistence in lymphocytes, likely including the use of miRNA transcription.

In contrast to mRNA gene expression, we found very few differentially expressed hosts or viral miRNAs in this study. Five significantly differentially expressed miRNAs were detected, two of which had human orthologs identified. Target prediction was only successful for the two miRNAs with human orthologs. Of those, one was upregulated (*eca-miR-652*) and predicted targets for differentially regulated genes in our samples for this miRNA included *TNRC6A*, a regulatory gene involved in gene silencing and gene expression regulation; *NPTN,* which plays an important role in cell adhesion molecule binding; *KPNA1,* which has been shown to play a role for nuclear import of proteins, including herpesvirus proteins [32], and *TP53,* which plays a role in the regulation of apoptosis and cell cycle regulation. Finally, in humans, *hsa-miR-652-3p* has been shown to interact with the endothelial repair gene *CCND2* and contributes to endothelial cell damage [33]. The other differentially regulated equine miRNA with the human ortholog *hsa-miR-6852-5p* was downregulated in response to infection. Interestingly, this miRNA is predicted to target a number of the differentially regulated identified target genes in our study which belong to the interferon-stimulated genes and are important components in the defense of herpesviruses. Further differentially regulated targets included the C1R ligand, which is part of the complement activation cascade; *CCL8,* an important chemokine, and a number of genes involved in apoptosis, cortisol metabolism, and cell adhesion. Even more intriguing is the fact that *hsa-miR-6852-5p* is predicted to bind tetraspanins (*TSPN9*, *TSPN11*, *TSPN14*, *TSPN15*, *TSPN17*, *TSPN18*, and *TSPN31*) as well as many collagens. A recent paper comparing horses that developed clinical EHM with horses that did not, identified a mutation in *TSPAN9* in horses with EHM in a genome-wide association study [35]. *TSPAN9* is expressed in endothelial cells and platelets and stabilizes the collagen receptor in platelet microdomains [36]. While the collagen receptor plays only a minor role in hemostasis, it is important in arterial thrombosis, ischemic stroke, and maintaining vascular integrity during inflammation. Interestingly, we have recently found that plasma collected from horses with clinical EHM responded with decreased aggregation in response to collagen (unpublished data). Furthermore, absence of *TSPAN9* in mice reduced collagen-induced activation and secretion [37]. While none of the predicted *TSPAN* or collagen genes were identified as differentially regulated in our samples, our study used PBMCs, which do not contain platelets or endothelial cells. However, both platelets and vascular endothelial cells are in direct contact with EHV-1-infected PBMCs during EHM pathogenesis and this interaction is likely an important aspect of the EHM pathogenesis. Interestingly, the human ortholog *hsa-miR-652-3p* is known to contribute to endothelial damage, and knockdown of this miRNA has been shown to promote endothelial cell repair [33]. Clearly, further investigation of this miRNA, its predicted targets in horses, and its role in EHM development in different tissue compartments would be interesting.

In conclusion, we characterized the transcriptome of equine PBMCs pre- and post-EHV-1 challenge infection using a repeated measure design. We used RNA deep sequencing to comprehensively investigate equine mRNA and miRNA transcripts, as well as those from four common equine herpesviruses, EHV-1, EHV-2, EHV-4, and EHV-5. Notably, after gene expression analysis, we were able to characterize the immune response of PBMCs during viremia and found significant upregulation of multiple genes involved in processes including the interferon pathway, T cell chemotaxis, and protein activation cascades such as the complement and coagulation system. More work is needed to fully elucidate how these mechanisms are involved in either protection from or contribution to EHV-1-associated secondary diseases such as abortion and EHM. Nevertheless, this work expands our current knowledge about the role of PBMCs during EHV-1 viremia and will inform the focus on future experiments to identify host and viral factors that contribute to clinical EHM.

## 4. Materials and Methods

### 4.1. Viruses

EHV-1 strain Ab4 (NCBI RefSeq: NC_001491.2) was propagated in equine dermal (NBL-6) cells (ATCC^®^ CCL-57™) with MEM-10 (Minimum Essential Medium Eagle (Sigma-Aldrich, St. Louis, MO, USA) supplemented with 100 IU/mL penicillin, 100 µg/mL streptomycin, 1% GlutaMAX (GIBCO, Life Technologies, Carlsbad, CA, USA), 1 mM sodium pyruvate, 1% non-essential amino acids (M7145, Sigma-Aldrich), and 10% fetal bovine serum. After incubation at 37 °C and 5% CO_2_ for 3–4 days, the cells were frozen and thawed, and cellular debris was removed by centrifugation at 300× *g* for 10 min. The stock was stored at −80 °C. Prior to inoculation in horses, the stock was thawed and sonicated for three cycles of 30 s at 50-1 amplification and diluted to a titer of 10^6^ plaque-forming units (PFU)/mL for inoculation of horses.

### 4.2. Animals

Seven 1-year-old horses (5 males, 2 females) were used in this experiment. Horses were screened prior to inclusion in the study to ensure virus neutralization assay (VN) blood serum titers were below 1:4 for EHV-1 and below 1:24 for EHV-4. Animals were housed in a building with natural ventilation with multiple horses per pen and nose-to-nose contact between pens. The horses had access to grass hay and water ad libitum for the entirety of the study. All animal maintenance and procedures were performed in compliance with Michigan State University’s Institutional Animal Care and Use Committee.

### 4.3. Experiment Design

The horses used in this study were part of a separate experiment evaluating the use of a human adenovirus vector expressing the EHV-1 inhibitory gene, IR2, for prevention of nasal viral shedding [27]. In this study, horses received intranasal instillation of either 3 × 10^10^ particles of a human adenovirus vector expressing the EHV-1 IR2 protein (*n* = 3; 2 males, 1 female) or 1.5 × 10^10^ particles of a null adenovirus vector (*n* = 4; 3 males, 1 female) by intranasal instillation two days prior to EHV-1 challenge [27]. Because no effect was observed as a result of IR2 treatment in any of the clinical, virological, or immunological parameters evaluated, for the purpose of the current study, all horses were considered as one group [27]. Horses were challenged with 5 × 10^7^ plaque-forming units (PFU) of EHV-1 Ab4 via intranasal instillation.

### 4.4. Sample Collection

To ensure we had samples for each horse’s day of peak viremia, 100 mL of whole blood was collected into heparinized syringes via jugular venipuncture from all horses 13 days prior to experimental infection with EHV-1 and daily for 10 days post-challenge infection. Samples were immediately transported back to the laboratory for PBMC isolation. For isolation of RNA and library preparation, PBMCs were separated by density gradient centrifugation over Histopaque-1077 (Sigma-Aldrich) as previously described [20] and cell pellets of 6 × 10^7^ were stored at −80 °C until RNA isolation. An additional aliquot of 1 × 10^7^ PBMCs was stored at −80 °C for quantification of viremia using qPCR as previously described [21]. The day after challenge for peak viral load in PBMCs was determined for each horse. Additionally, physical exams to evaluate respiratory disease and rectal body temperature were performed daily and nasal swabs were collected for viral quantification using qPCR, as previously described [21].

### 4.5. RNA Isolation, Library Preparation, and Sequencing

A summary of the data analysis can be seen in Figure 8. RNA for RNA sequencing analysis was isolated from PBMCs from each horse pre-challenge, as well as on the day of peak viremia. Cell pellets were lysed and homogenized using TRIzol Reagent (Thermo Fisher) following the manufacturer’s instructions. The aqueous phase was then collected, washed with 100% ethanol, and total RNA was isolated using the miRNeasy Mini Kit (Qiagen) according to the manufacturer’s instructions. To eliminate genomic DNA contamination, a deoxyribonuclease treatment (Qiagen) was applied to each sample according to the manufacturer’s recommendation. The concentration of RNA was determined using fluorometric quantification with Qubit 1.0. RNA quality was evaluated using the Agilent 2100 Bioanalyzer with the RNA 6000 Pico Assay, and samples with an RNA integrity number (RIN) score ≥ 8.90 were submitted for sequencing. Library preparation and next-generation sequencing were performed at Michigan State University’s Genomics Research and Technology Support Facility. Briefly, stranded mRNA cDNA library preparation was preformed using the Illumina TruSeq Stranded mRNA kit with Integrated DNA Technologies (IDT) for Illumina Unique Dual Index adapters according to the manufacturer’s recommendations. miRNA cDNA libraries were prepared using the Illumina TruSeq Small RNA Library Preparation Kit following the manufacturer’s recommendations. Completed libraries were quality-controlled and quantified using a combination of Qubit dsDNA HS and Advanced Analytical Fragment Analyzer High Sensitivity DNA assays. The mRNA libraries were pooled in equimolar quantities for sequencing and the pool was quantified using the Kapa Biosystems Illumina Library Quantification qPCR kit. This pool was loaded onto two (2) lanes of an Illumina HiSeq 4000 flow cell and sequencing was performed in a 2 × 150 bp paired-end format using HiSeq 4000 styrene-butadiene-styrene (SBS) reagents. The small libraries were pooled in equimolar amounts for multiplexed sequencing and the pool was quantified using the Kapa Biosystems Illumina Library Quantification qPCR kit. This pool was loaded onto one (1) lane of an Illumina HiSeq 4000 flow cell and sequencing was performed in a 1 × 50 bp single read format using HiSeq 4000 SBS reagents. Base calling was done using Illumina Real-Time Analysis (RTA) v2.7.7 and output of RTA was demultiplexed and converted to FastQ format with Illumina Bcl2fastq v2.19.1. All raw sequencing reads are available in the National Center for Biotechnology Information (NCBI) sequence read archive (SRA) under BioProject Ascension number PRJNA681404.

### 4.6. Genome-Guided mRNA Alignment

Read quality was assessed using FastQC software v0.11.7. Reads were mapped to the *Equus caballus* genome (assembly EquCab3.0, ENSEMBL release-95) using HISAT2v 2.1.0. with these options: --rna-strandness RF --dta. The accepted hits’ BAM files were sorted by name using SAMtools v1.5 and read counts were generated using htseq-count (built in with Python v3.6.4) with the following options: --format=bam, --stranded = reverse, and –order = name [98,99,100]. The reference annotation GTF file was downloaded from Ensembl release 95.

### 4.7. Host and Viral miRNA Identification and Quantification

Read quality was assessed before and after quality and adaptor trimming using FastQC software v0.11.7 [101]. Raw reads were trimmed using cutadapt v1.16. Options included trimming the Illumina adaptor sequence (option -a TGGAATTCTCGGGTGCCAAGG), reads shorter than 15 bp were discarded (option -m 15), and the 3′ end was trimmed with a quality score cutoff of 20 (option -q 20) [102]. After trimming, miRDeep2 v2.0.0.8 was used to identify all miRNAs present in the samples, including putative novel miRNAs as well as known miRNAs [103]. For this, reads from all samples were first pooled. Next, a combined reference genome was created from combining the horse (EquCab3.0; downloaded from Ensembl release 95) with the genome of the four most common equine herpesviruses (EHV-1, NCBI RefSeq NC_001491.2; EHV-2, NCBI RefSeq NC_001650.2; EHV-4, NCBI RefSeq NC_001844.1; and EHV-5, NCBI RefSeq NC_026421.1) and indexed using the bowtie-build function of Bowtie v1.2.2. The mapper.pl function of miRDeep2 was used on the pooled reads to the reference genome to create a fasta file with processed reads and an arf file with mapped reads. Next, the miRDeep2.pl function was performed on these outputs using equine miRNAs as the main reference. Since the equine miRNA database is still very incomplete, mouse and human known miRNAs were used as related species lists. Known equine, mouse, and human miRNAs were downloaded from the miRbase database (release 22.1) [104]. Novel RNAs with an miRDeep score <1 were removed for subsequent analysis.

In order to perform differential gene expression analysis, the miRNAs were quantified in each sample. For this, the quantifier.pl tool from the miRDeep2 package was used. For this, all novel mature and precursor miRNAs were added to the list of known miRNAs from horse, mouse, and human. Quantifier.pl was run with the processed reads fasta file as input and the -k option was used to consider precursor mature mappings that have different IDs.

### 4.8. Differential Gene Expression

Differential gene expression analyses for both mRNA gene counts and miRNA counts were performed using the DESeq2 package v1.22.2 in R v3.5.3 [105]. The design formula used was “~horse + timepoint” so that the variable of interest was “timepoint” (pre-infection vs. post-infection) and controlling for the effect of “horse”. Genes or miRNAs with less than 5 total read counts were filtered out prior to analysis. The default DESeq2 analysis was performed using the DESeq function. p-values were adjusted using the Benjamini–Hochberg method (the default for DESeq2). Significance was set at adjusted p-value (padj) <0.05 and log2 fold change >|3| for mRNA and padj <0.05 and log2 fold change >|1| for miRNA.

### 4.9. Gene Enrichment Analysis

A gene enrichment analysis for gene ontology (GO) terms was performed on the gene lists derived from the differential expression analysis. Horse Ensembl IDs were converted to gene symbols based on the gene names identified in the EquCab3.0 GTF file (Ensembl release 95). A GO enrichment analysis for biological processes was performed using the enrichGO function from the clusterProfiler package [106]. The database used was the human “org.Hs.eg.db”, and the background was created from the gene symbols associated with all genes present in our samples [107]. p-values were adjusted using the Benjamini–Hochberg correction and statistical significance was set at *p* < 0.05. After generating lists of enriched GO terms, redundant terms were removed using REVIGO [108]. Enriched GO terms along with their associated adjusted p-values were provided as inputs, and the allowed similarity was small (0.5). The default settings were used, which included selecting the whole UniProt database to determine GO term sizes and using the SimRel semantic similarity measure for the analysis.

### 4.10. Target Gene Analysis of Differentially Expressed miRNAs

A target gene analysis of the 2 differentially expressed miRNAs with an identified human ortholog was conducted using TargetScan Release v.7.2 [109]. The other identified differentially expressed miRNAs did not result in any predicted target genes as no human ortholog could be identified with our analysis. Predicted target genes for miRNAs with human orthologs were compared with the differentially expressed gene list (Table A1 in Appendix B).

### 4.11. Viral Gene Identification and Counting

In order to identify viral reads within the samples, a combined reference genome was created from combining the horse (EquCab3.0; Ensembl release 95) with the genome of the four most commonly studied equine herpesviruses (EHV-1, NCBI RefSeq NC_001491.2; EHV-2, NCBI RefSeq NC_001650.2; EHV-4, NCBI RefSeq NC_001844.1; and EHV-5, NCBI RefSeq NC_026421.1). This allowed for subsequent normalization of viral read counts to consider total gene expression from the host. Alignment to this genome was performed for each sample using HISAT2, as described above. Gene counts were then generated using StringTie v1.3.5 as described in the StringTie manual (accessed June 2020) [100]. Initially, gene counts were generated for each sample using the equine reference GTF file. Then, a new annotation in GTF format was generated based on the genes and transcripts present in our samples. Finally, the BAM file for each sample was re-run with StringTie to generate gene counts for each sample using the new merged GTF file as the reference annotation and included the -B and -e options. Transcripts per million (TPM) values for each gene were extracted from the gene abundance files generated using the -A option. Raw read coverages of samples on the EHV-1 genome were visualized using Integrative Genomics Viewer (IGV) v2.4.10 [110].

### 4.12. In Silico Cell Sorting

Normalized read counts expressed as counts per million (CPM) were determined using EdgeR v3.24.3 and values of redundant genes were merged together [111]. Cell fractions were imputed using CIBERSORTx [28]. The included LM22 (22 immune cell types based on human immune cell gene signatures) was used as the signature matrix file. The mixture file included the CPM values for each gene from all samples, identified with gene symbol. B-mode batch correction was performed, and quantile normalization was disabled (recommended for RNA sequencing data). Permutations for significance analysis were set at 100. The run was performed in relative mode (default). Each cell type was represented as a percentage of total cells for each sample, and a Wilcoxon signed rank test was performed on the percentages for each cell type using R.

## Figures and Tables

**Figure 1 pathogens-10-00043-f001:**
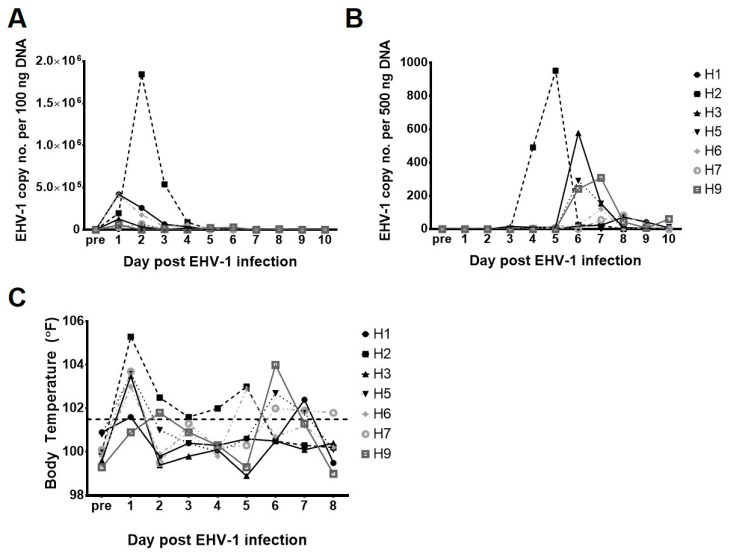
Clinical and virological disease post equine herpesvirus 1 (EHV-1) challenge. (**A**) EHV-1 nasal shedding. Data are expressed as EHV-1 copy number per 100 ng template DNA as determined by qPCR. (**B**) Viremia. Data are expressed as EHV-1 copy number per 500 ng template DNA as determined by qPCR. (**C**) Body temperature. A body temperature over 101.5 °F was considered as a fever and is indicated by the horizontal dashed line.

**Figure 2 pathogens-10-00043-f002:**
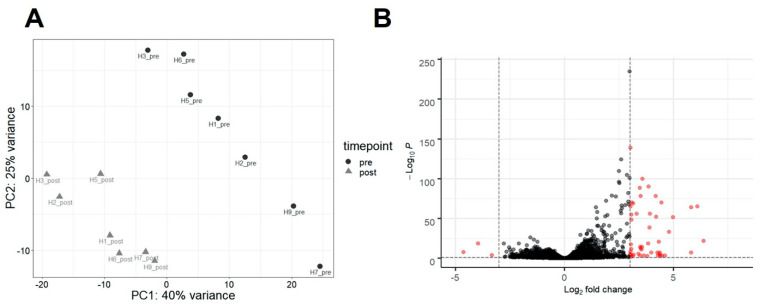
(**A**) Principal component analysis. Samples from horses pre-infection (black dots) cluster separately from samples during viremia (gray triangles). The horse numbers are indicated on the plot as H1–H9, and H3 and H5 were females. (**B**) Differentially expressed genes in horses pre-infection vs. post-infection. Genes with positive log2 fold change (x-axis) indicate genes upregulated in peripheral blood mononuclear cells (PBMCs) during EHV-1 infection compared to pre-infection while negative log2 fold change values indicate genes downregulated in PBMCs during infection. p-values are expressed on the y-axis, with more significantly differentially expressed genes towards the top of the plot. Genes highlighted in red passed the threshold of significance set at adjusted *p*-value < 0.05 and log2 fold change > |3|.

**Figure 3 pathogens-10-00043-f003:**
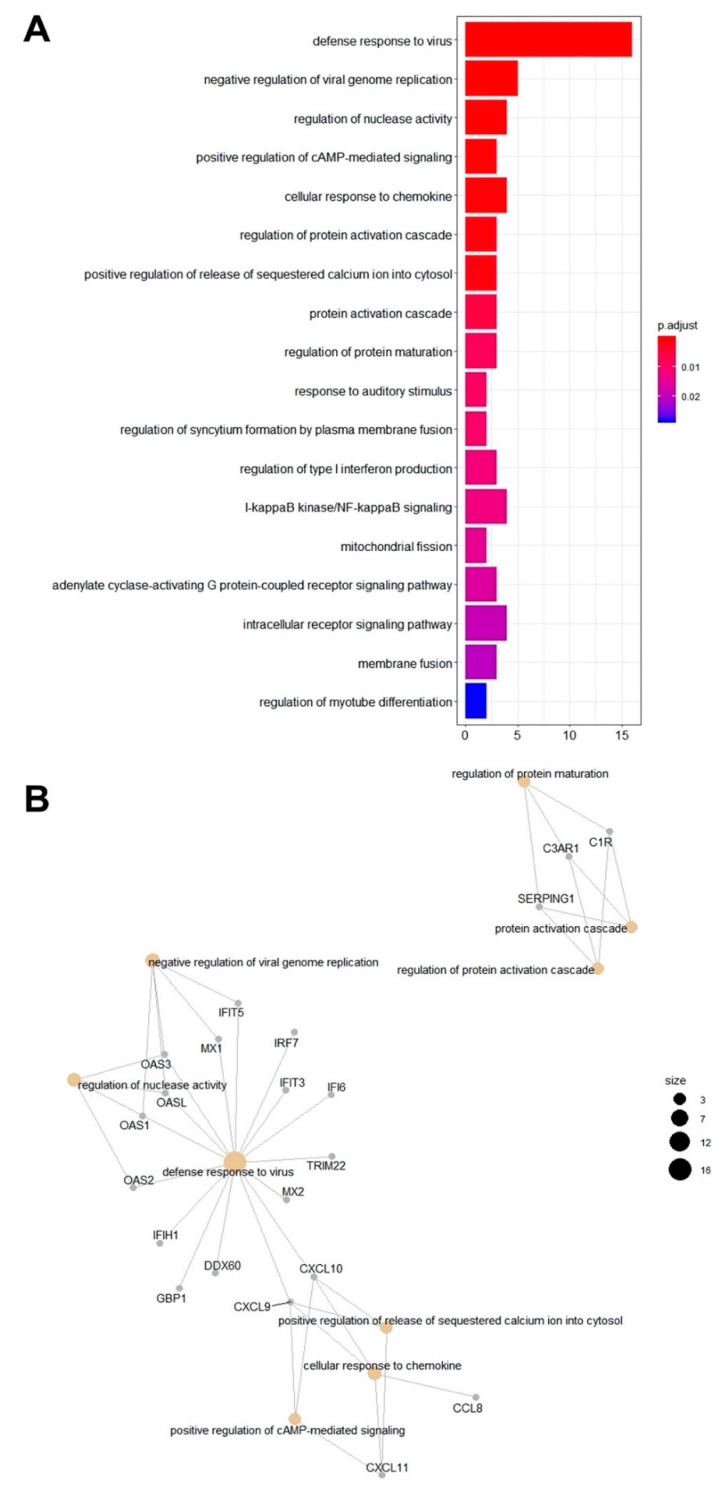
(**A**) Gene ontology (GO) terms for biological processes overrepresented in horses during EHV-1 infection. GO term enrichment analysis was performed using the enrichgo function of the clusterprofiler package in R. The resulting terms were filtered for redundancy using REVIGO. The 18 non-redundant enriched GO terms are visualized here. The most significantly enriched terms are at the top and are listed in decreasing significance (increasing padj). The number of genes from our gene list are indicated on the x-axis. (**B**) Net plot of the most significantly enriched GO terms and associated genes. The ten non-redundant GO terms with the lowest padj values are listed here with the associated genes from our gene list. Tan nodes represent the GO term and gray nodes represent genes. The size of the GO term nodes indicates the number of genes from our list associated with that term. The biological processes and the associated genes cluster based on similarity.

**Figure 4 pathogens-10-00043-f004:**
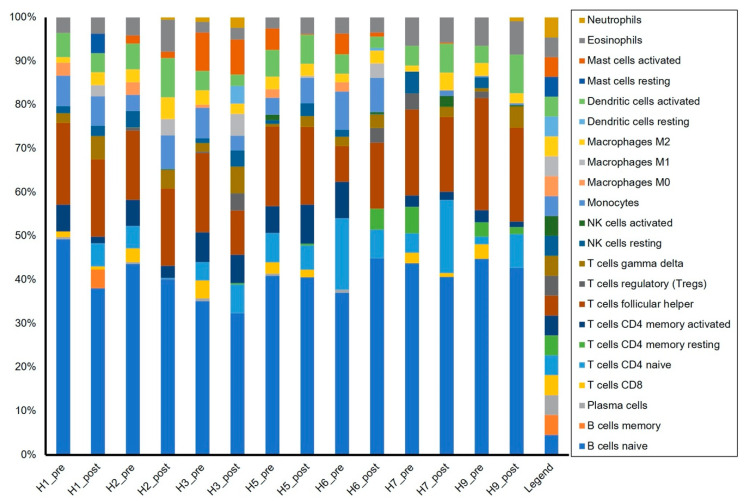
Cell population fractions in PBMC samples. Cell population fractions for each sample were estimated using CIBERSORTx and the reference gene signature “LM22” included with the software, which is based on the transcriptome of human PBMC samples with pre-determined cell populations. Twenty-two cell subpopulations are represented by different colors. Fractions are expressed as percent of the total population (y-axis) for each sample (x-axis).

**Figure 5 pathogens-10-00043-f005:**
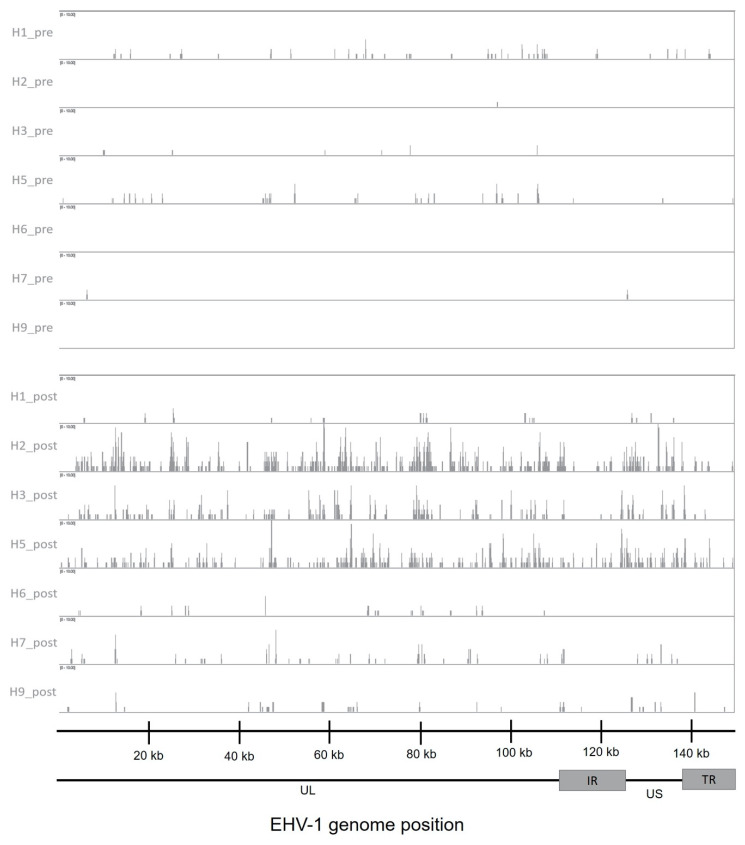
Read coverage plot of the EHV-1 genome. The top seven tracks are the pre-infection samples, and the bottom seven tracks are post-infection samples. Reads were aligned to the EHV-1 strain Ab4 genome (NCBI RefSeq NC_001491.2). UL = unique long region; IR = internal repeat region; US = unique small region; TR = terminal repeat region, as described in [30].

**Figure 6 pathogens-10-00043-f006:**
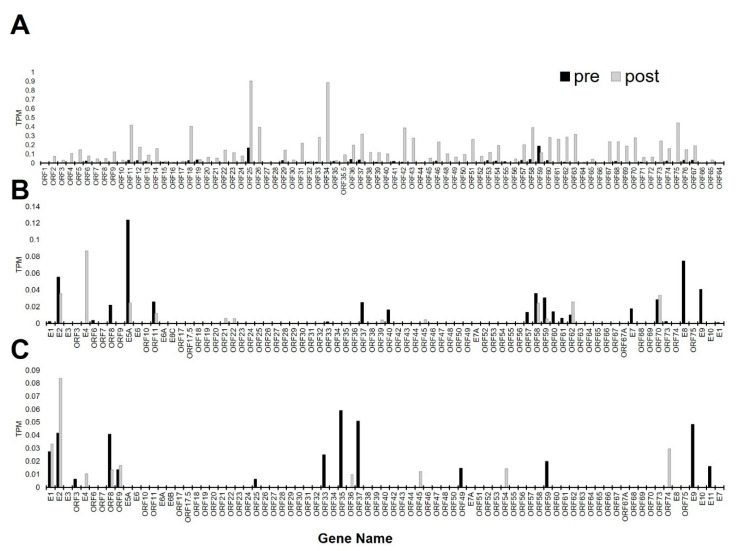
Normalized counts of viral genes. Data represent the average transcripts per million (TPM) of EHV-1 (**A**), EHV-2 (**B**), and EHV-5 (**C**). The gray line is pre-EHV-1 challenge infection, and the black line is post-EHV-1 challenge.

**Figure 7 pathogens-10-00043-f007:**
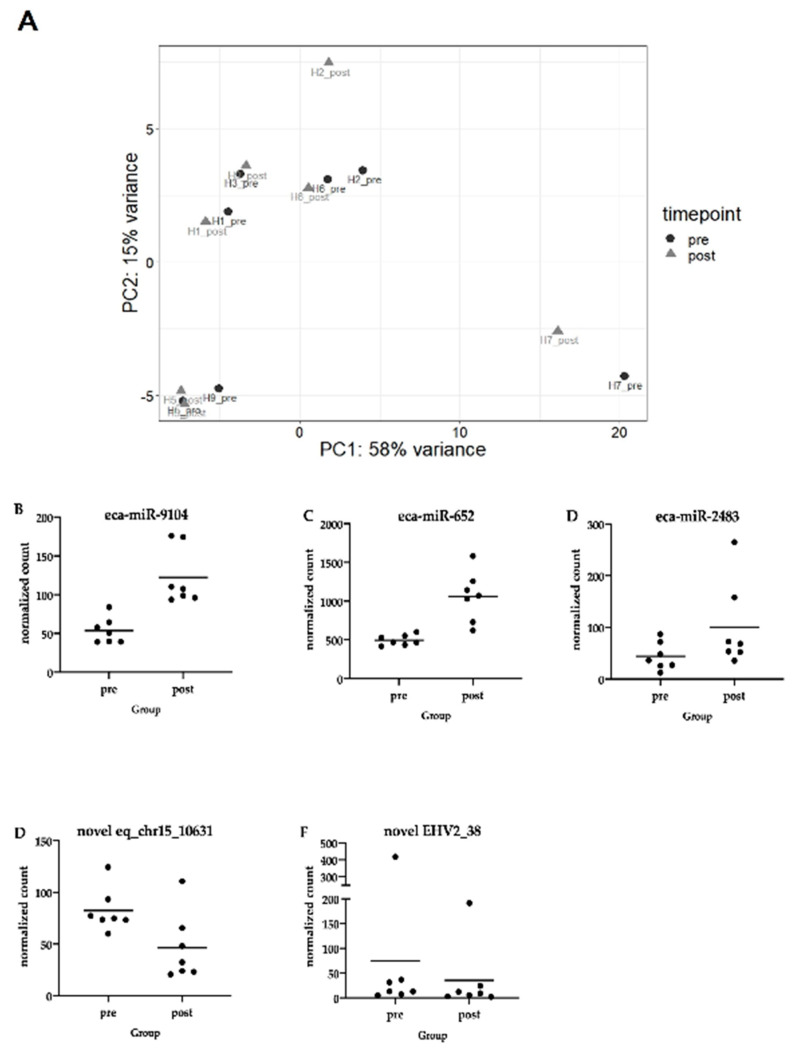
(**A**) Principal component analysis of miRNA samples. Samples cluster based on horse rather than from infection status pre-infection (black dot) vs. during infection (gray triangle). The horse numbers are indicated on the plot as H1–H9, and H3 and H5 were females. Normalized read counts of differentially expressed miRNAs. (**B**) eq_chr11_2567 identified with *eca-miR-9104*; (**C**) eq_chrx_44985 identified with eca-miR-652 and the human ortholog hsa-miR-652-3p; (**D**) eq_chrx_45803 identified with eca-miR-2483; (**E**) eq_chr15_10631 identified with the human ortholog hsa-miR-6852-5p; (**F**) EHV-2_38 aligned to the EHV-2 viral genome. Details regarding these miRNAs can be found in Table 3.

**Figure 8 pathogens-10-00043-f008:**
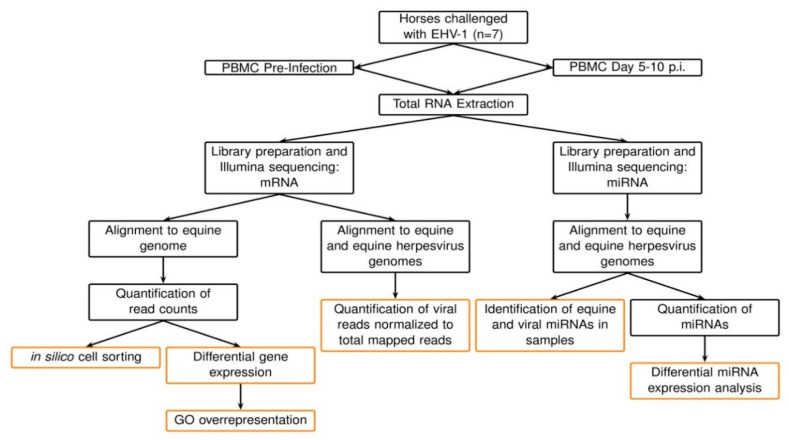
Summary of data analysis. RNA was extracted from PBMCs from horses prior to and during EHV-1 challenge infection. Total RNA was processed with next-generation RNA sequencing for mRNA and miRNA. Black boxes represent upstream data analysis processes, while the red boxes represent processes that resulted in output reported in the results.

**Table 1 pathogens-10-00043-t001:** Mapping summary statistics of mRNA sequencing.

Sample ID	Total Reads	Uniquely Mapped (Number of Reads)	Uniquely Mapped (%)
H1_PRE	26,999,513	21,296,243	78.9
H1_POST	27,536,037	20,976,540	76.2
H2_PRE	35,256,030	29,155,433	82.7
H2_POST	42,517,852	33,582,323	79.0
H3_PRE	56,397,724	45,505,043	80.7
H3_POST	63,282,694	51,758,322	81.8
H5_PRE	93,180,427	75,568,952	81.1
H5_POST	27,862,507	23,309,213	83.7
H6_PRE	50,453,360	40,889,209	81.0
H6_POST	24,740,083	19,062,322	77.1
H7_PRE	26,710,306	20,926,895	78.3
H7_POST	52,898,636	43,121,055	81.5
H9_PRE	51,887,615	42,716,415	82.3
H9_POST	28,998,398	23,182,557	79.9
Average	43,480,084	35,075,037	80.3

Mapping summary statistics of mRNA sequencing. Total reads after sequencing and the number and percent that uniquely mapped to the EquCab3.0 genome are shown for each sample.

**Table 2 pathogens-10-00043-t002:** Average fraction of cell populations.

	Pre-Challenge (% of Total Cell Population)	Post-Challenge (% of Total Cell Population)
B cells naive	42.07 ± 0.02	39.89 ± 0.01
B cells memory	0.00 ± 0.00	0.62 ± 0.01
Plasma cells	0.38 ± 0.00	0.00 ± 0.00 *
T cells CD8	2.37 ± 0.00	0.46 ± 0.00 **
T cells CD4 naive	5.52 ± 0.02	6.95 ± 0.02
T cells CD4 memory resting	1.34 ± 0.01	1.04 ± 0.01
T cells CD4 memory activated	5.55 ± 0.01	3.28 ± 0.01
T cells follicular helper	17.75 ± 0.02	16.69 ± 0.01
T cells regulatory (Tregs)	0.88 ± 0.00	1.05 ± 0.01
T cells gamma delta	1.11 ± 0.00	4.00 ± 0.00 **
Natural killer cells resting	2.32 ± 0.01	1.32 ± 0.01
Natural killer cells activated	0.18 ± 0.00	0.51 ± 0.00
Monocytes	4.38 ± 0.01	4.71 ± 0.01
Macrophages M0	1.49 ± 0.00	0.00 ± 0.00 *
Macrophages M1	0.00 ± 0.00	2.14 ± 0.01 *
Macrophages M2	2.39 ± 0.00	3.18 ± 0.00
Dendritic cells resting	0.00 ± 0.00	0.66 ± 0.00
Dendritic cells activated	4.98 ± 0.00	5.84 ± 0.01
Mast cells resting	0.00 ± 0.00	0.64 ± 0.01
Mast cells activated	2.89 ± 0.01	1.54 ± 0.01
Eosinophils	4.22 ± 0.01	4.92 ± 0.01
Neutrophils	0.15 ± 0.00	0.55 ± 0.00

The average fraction of cell population fractions for each sample was estimated using CIBERSORTx [28] and the reference gene signature “LM22” included with the software, which is based off of the transcriptome of human PBMC samples with pre-determined cell populations. Wilcoxon’s signed-rank test was performed. * indicates *p* ≤ 0.1 and ** indicates *p* ≤ 0.05.

**Table 3 pathogens-10-00043-t003:** **Differentially expressed miRNAs and common differentially expressed genes (predicted) to be targeted.** Differentially expressed miRNAs were determined by adjusting the *p*-value < 0.05 and log 2 fold change >|1|. “Upregulated” refers to miRNAs upregulated during viremia compared to pre-challenge infection and “downregulated” refers to miRNAs downregulated during viremia compared to pre-challenge infection. Predicted genes to be targeted by miRNAs during viremia include only genes to be found significantly differentially regulated in our gene expression analysis (see also Table A1). A significant upregulation during viremia of those genes is indicated by ↑ and a significant downregulation is indicated by ↓.

ID	Log2 Fold	Padj	MirBase ID	Human Ortholog	Mature Sequence	Precursor Sequence	Differentially Expressed Gene Targets
	**Upregulated miRNAs**						
Equine_chr11_2567	1.2	3.5 × 10^−8^	eca-miR-9104		CTGACCTGAGGCCTCTGCTGCA	GAGTGGCTGGGCTCAGCAGGGCGGAGGGTCAGGAGGTGAGCTTGGCTCTGCTGACCTGAGGCCTCTGCTGCA	
Equine_chrX_44985	1.1	1.3 × 10^−17^	eca-miR-652	hsa-miR-652-3p	AATGGCGCCACTAGGGTTG	CAACCCTAGGAGAGGGTGCCATTCACATAGACTATAATTGAATGGCGCCACTAGGGTTG	TNRC6A↓, NPTN↑, KPNA1↓, TP53↑
Equine_chrX_45803	1.1	2.0 × 10^−3^	eca-miR-2483		TCTGTCAACCATCCAGCTGTTT	TCTGTCAACCATCCAGCTGTTTGGGGTGATGCAAACAAACATCTAGTTGGTTGAGAGAAT	
	**Downregulated miRNAs**						
Equine_chr15_10631	−1	1.4 × 10^−2^	-	hsa-miR-6852-5p	ACCTGGGGATCTGAGGAG	ACCTGGGGATCTGAGGAGGCCCTTCCAGCCCCAAGGCTGGGAATGCTCCTGGTCCCCTTTCTTGC	IFIT5↑, Mx2↑, OAS3↑, OAS2↑, CCL8↑, BCL2LI4↑, MPZ↑, TGM2↑, HSDIIB1↑ FN1↓, DEFB1↓
EHV-2_38	−1	1.0 × 10^−2^	-		TATGATAGTCCATACCCTTAAGT	TATGATAGTCCATACCCTTAAGTTTGATAAGTAAAAAATTTAAGTACGTGGACTGTCAACA	

## Data Availability

Raw sequencing reads used for the data analysis in this study are available in the National Center for Biotechnology Information (NCBI) sequence read archive (SRA) under BioProject Ascension number PRJNA681404.

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
