# Peer review of "Transcriptomic Profiling of Equine and Viral Genes in Peripheral Blood Mononuclear Cells in Horses during Equine Herpesvirus 1 Infection"

_pathogens, 2021, doi:10.3390/pathogens10010043_

Round 1
Reviewer 1 Report
Dear Authors,
the aim of your study was to establish a gene expression analysis of host and equine herpesvirus genes during EHV-1 viremia using RNA sequencing. Potentially novel miRNAs that mapped to the EHV-2 and EHV-5 genomes were identified. Of those, 4 equine and 1 EHV-5 miRNAs were differentially expressed in PBMCs during viremia. This work expands the knowledge about the role of PBMCs during EHV viremia and will inform the focus on future experiments to identify host and viral factors that contribute to clinical equine diseases due to herpesvirus.
The sections of the article are well written and deeply described.
It could be useful to include more literature regarding EHV-5 (Lines 423-463, line 432 partcularly) and its ability to be latent and to induce challenges for the systemic immunity
1)Miglio, Arianna, Morelli, Chiara, Gialletti, Rodolfo, Lauteri, Eleonora, Sforna, Monica, Marenzoni, Maria Luisa, Antognoni, Maria Teresa (2019). Clinical and immunophenotypic findings in 4 forms of equine lymphoma. CANADIAN VETERINARY JOURNAL, vol. 60, p. 33-40, ISSN: 0008-5286
2) Miglio, Arianna, Antognoni, Maria Teresa, Morelli, Chiara, Gialletti, Rodolfo (2018).Third Eyelid T-cell-Rich Large B-cell Lymphoma Positive to EHV-5 in a Mare—a Case Report. JOURNAL OF EQUINE VETERINARY SCIENCE, vol. 70, p. 52-56, ISSN: 0737-0806, doi: 10.1016/j.jevs.2018.08.007
Your article is really interestingly and only minor revisions should be added, including the English revision by an expert.
Author Response
Reviewer 1:
Dear Authors,
the aim of your study was to establish a gene expression analysis of host and equine herpesvirus genes during EHV-1 viremia using RNA sequencing. Potentially novel miRNAs that mapped to the EHV-2 and EHV-5 genomes were identified. Of those, 4 equine and 1 EHV-5 miRNAs were differentially expressed in PBMCs during viremia. This work expands the knowledge about the role of PBMCs during EHV viremia and will inform the focus on future experiments to identify host and viral factors that contribute to clinical equine diseases due to herpesvirus.
The sections of the article are well written and deeply described.
It could be useful to include more literature regarding EHV-5 (Lines 423-463, line 432 partcularly) and its ability to be latent and to induce challenges for the systemic immunity
1)Miglio, Arianna, Morelli, Chiara, Gialletti, Rodolfo, Lauteri, Eleonora, Sforna, Monica, Marenzoni, Maria Luisa, Antognoni, Maria Teresa (2019). Clinical and immunophenotypic findings in 4 forms of equine lymphoma. CANADIAN VETERINARY JOURNAL, vol. 60, p. 33-40, ISSN: 0008-5286
2) Miglio, Arianna, Antognoni, Maria Teresa, Morelli, Chiara, Gialletti, Rodolfo (2018).Third Eyelid T-cell-Rich Large B-cell Lymphoma Positive to EHV-5 in a Mare—a Case Report. JOURNAL OF EQUINE VETERINARY SCIENCE, vol. 70, p. 52-56, ISSN: 0737-0806, doi: 10.1016/j.jevs.2018.08.007
More information and references were added in the manuscript (lines 427-438) and the changes in the document are indicated in red.
Your article is really interestingly and only minor revisions should be added, including the English revision by an expert.
A further revision in terms of grammar/spelling/and typos has been performed and all changes are indicated in red in the main document.
Reviewer 2 Report
The article is really well done. The experimental design is perfectly clear and well write. Without any doubt, the results will contribute to understand the mechanisms of disease. Congratulations for such a good paper.
Author Response
Thank you, we appreciate the comment.
Reviewer 3 Report
The following is a review of the manuscript “Transcriptomic profiling of equine and viral genes in peripheral blood mononuclear cells in horses during equine herpesvirus1 infection by Zarski et. al submitted to MDPI Pathogens for consideration. In this manuscript the authors characterize the mRNA and microRNA expression profiles of Equine herpesivirus 1 infected equine PBMCs to identify acute phase signatures. They identify 51 differentially expressed genes, viral transcripts and host encoded and virus encoded microRNAs many of which were found to exhibit differential expression pre- and post-infection suggesting that they may have roles in the regulation of the immune response to infection. Overall, the authors must be complimented for a well written manuscript that also describes the methods and materials in sufficient details to enable reproduction. Data typically support the conclusions drawn by the authors. Listed below are the areas of improvement for this manuscript.
- Among the 7 infected horses, two were female. Did the authors observe any gender correlations in the immune response to infection?
- Horse H6 died of EHM; yet shows the least abundance of EHV1 transcripts post infection. Please explain this anomaly.
- Please elaborate on why different quantities (100ng vs 500ng) of DNA were used to enumerate EHV1 copy numbers between Figure 1a and 1b. I assume that shedding was more prominent compared to viraemia, but this is not clear in the text and would improve the manuscript.
- Did the authors check for any co-infections in these animals after the onset of EHV1 disease ?
- The study identifies a high level of EHV1 viral transcription, gene transcription yet does not identify any EHV-1 miRNAs but rather EHV-2 and EHV-5 miRNAs in the dataset. This is intriguing. Please speculate if gamma and alpha Herpesviridae infections are mutually exclusive because of selective viral miRNA expression.
- Table 3 and Figure 7A should be moved to supplementary methods. Instead, include miR target data (Table A2) as part of the main manuscript highlighting only those genes that are targeted by the miRs in 7b, c,d e and f. Identify common genes (predicted) to be targeted and find correlations with the gene expression data from your analysis to identify inverse correlations.
- In Figure 7b-f, please show medians of scatterplots and indicate if differences are statistically significant.
- Line 244. The authors identify that miR counts vary more between horses than with within infection groups. Is there any correlation with age or degree of viraemia? This should be elaborated further. What was the miR expression status in horse 6?
- One interesting follow up on this study should be address which cells express the differentially expressed miRNAs identified. If the authors have the sorted cell populations, this would be a relatively easy to measure for the three miRs in Figure 7.
Author Response
The following is a review of the manuscript “Transcriptomic profiling of equine and viral genes in peripheral blood mononuclear cells in horses during equine herpesvirus1 infection by Zarski et. al submitted to MDPI Pathogens for consideration. In this manuscript the authors characterize the mRNA and microRNA expression profiles of Equine herpesivirus 1 infected equine PBMCs to identify acute phase signatures. They identify 51 differentially expressed genes, viral transcripts and host encoded and virus encoded microRNAs many of which were found to exhibit differential expression pre- and post-infection suggesting that they may have roles in the regulation of the immune response to infection. Overall, the authors must be complimented for a well written manuscript that also describes the methods and materials in sufficient details to enable reproduction. Data typically support the conclusions drawn by the authors. Listed below are the areas of improvement for this manuscript.
- Among the 7 infected horses, two were female. Did the authors observe any gender correlations in the immune response to infection?
The question of gender is an important question but is not one that could be addressed in this study, as we only had 2 female horses which does not allow for a satisfactory statistical analysis. We did however add identifiers to the legends of the PCA plots and indicated which horses were female as to allow the reader to see how they clustered compared to the other horses.
- Horse H6 died of EHM; yet shows the least abundance of EHV1 transcripts post infection. Please explain this anomaly.
It is an interesting observation that horse 6 was not one of the horses with the highest levels of viremia. However, while viremia is essential for transport of the virus to the spinal cord vasculature, this is not an absolute correlation as the pathogenesis of EHM depends on multiple host and viral factors. Further, while it is true that there is no EHM in the absence of viremia, it is not entirely clear whether higher levels or longer duration of viremia and possibly the timing are more likely to lead to EHM. We have previously done multiple studies with the same strain and dose of virus where the horses with clinical EHM were not always the horses with the highest levels of viremia either (see also HOLZ at al. J. Gen. Virol. 2017, 98, doi:10.1099/jgv.0.000773.).
- Please elaborate on why different quantities (100ng vs 500ng) of DNA were used to enumerate EHV1 copy numbers between Figure 1a and 1b. I assume that shedding was more prominent compared to viraemia, but this is not clear in the text and would improve the manuscript.
We added a sentence in line 105 explaining that there typically is more viral DNA in the nasal secretions, than in the PBMCs. The analysis is common procedure and was actually part of a different manuscript (referenced in the present manuscript), which is why we did not elaborate further in this manuscript.
- Did the authors check for any co-infections in these animals after the onset of EHV1 disease ?
The horses are kept in an isolation facility for the duration of the study with no exposure to other horses or staff. We perform regular physical exams and test for EHV-1 and EHV-4 as part of our screening process for the horses. Testing for other viral infections or bacterial infection is only performed if there is an indication that those may have occurred based on clinical signs because the chances of this occurring are very low in our experimental environment.
- The study identifies a high level of EHV1 viral transcription, gene transcription yet does not identify any EHV-1 miRNAs but rather EHV-2 and EHV-5 miRNAs in the dataset. This is intriguing. Please speculate if gamma and alpha Herpesviridae infections are mutually exclusive because of selective viral miRNA expression.
Based on our study, there is no indication that co-infection with alpha and gamma herpesviruses is mutually exclusive as we are clearly observing both simultaneously. As indicated in the manuscript, detection of EHV-2 and EHV-5 in PBMCs is common in horses and likely related to the fact that B-cells are a site of latency for these viruses. For EHV-1 and EHV-4, the retropharyngeal lymph nodes and trigeminal ganglia are the presumed sites of latency, which may explain while we do not see regulatory miRNA expression in the PBMCs for these viruses. It would be interesting to study how the miRNA expression compares in other sites such as the lymph nodes or trigeminal ganglia during latency, but this was not the goal of this study.
- Table 3 and Figure 7A should be moved to supplementary methods. Instead, include miR target data (Table A2) as part of the main manuscript highlighting only those genes that are targeted by the miRs in 7b, c,d e and f. Identify common genes (predicted) to be targeted and find correlations with the gene expression data from your analysis to identify inverse correlations.
MIRDB.org
We switched table 3 and table 2a. As for figure 7a, we feel that this figure gives important information on how the individual horses clustered and allows to compare clustering of horses with highest viremia, females, or the EHM horse with the rest of the horses. Identifiers have been added to the legend to highlight this aspect.
A target gene analysis of the 2 differentially expressed miRNA with an identified human ortholog was done using TargetScan Release v.7.2 . The other identified differentially expressed miRNAs did not result in any predicted target genes as no human orthologue could be identified with our analysis. Predicted target genes for miRNAs with human orthologues were compared with the differentially expressed gene list and a number of genes were identified and added to the table and discussed. Further, arrows were added to the table to indicate whether the target genes were differentially up- or down-regulated. Details describing the results of the target gene analysis have been added to the results section and the discussion in line 263-289 and 481-511.
- In Figure 7b-f, please show medians of scatterplots and indicate if differences are statistically significant.
A mean was added to the figures as a visual aid. As for statistical analysis, this was done as part of the miRNA analysis and the shown miRNA all came up statistically significant in the analysis at an adjusted p<0.05.
- Line 244. The authors identify that miR counts vary more between horses than with within infection groups. Is there any correlation with age or degree of viraemia? This should be elaborated further. What was the miR expression status in horse 6?
All horses in this study were yearling horses and there was no correlation with degree of viremia and miRNA expression. Further, the EHM horse (H6) does not look different from the other horses in the PCA plot. Horse numbers are actually indicated in the PCA plot (H1-H9) as well as in the viremia graph and we have highlighted this in the figure legend of the PCA plot now. In addition, we added a sentence to the manuscript (line 246-248) to clarify.
- One interesting follow up on this study should be address which cells express the differentially expressed miRNAs identified. If the authors have the sorted cell populations, this would be a relatively easy to measure for the three miRs in Figure 7.
Unfortunately, this study did not use sorted cell populations but rather a program (CIBERSORTx), that estimates cell population fractions of each sample based on a reference gene signature. Thus, analysis of individually sorted populations for mRNA expression and miRNA would have to be a goal of a future study.